# Importance of Community Forestry Funds for Rural Development in Nepal

**Puspa K. C. Bhandari [1], Prabin Bhusal [1,*], Ganesh Paudel [2] , Chiranjibi P. Upadhyaya [1] and Bir Bahadur Khanal Chhetri [1]**

[1] Institute of Forestry, Tribhuvan University, Pokhara Campus, Pokhara 33700, Nepal;
puspa_kcbhandari@yahoo.co.uk (P.K.C.B.); cpupadhyaya@hotmail.com (C.P.U.);
bbkchhetri@iofpc.edu.np (B.B.K.C.)

[2] Ministry of Forests and Environment, Kathmandu 3987, Nepal; ecopaudel@gmail.com

[*] Correspondence: pbhusal@iofpc.edu.np; Tel.: +977-9849215453

**Abstract:** Nepal's Community Forestry (CF) process has implied the devolution of powers to collect, retain, and redistribute forest revenue from community forests products. This study contributes to our knowledge about these important aspects of CF by presenting an analysis of the dynamic pattern of income and expenditure of 43 randomly selected Community Forestry User Groups (CFUGs) from Kaski, Nepal. Results show that CFUG three-year average annual income accounts NRs 216,225 (1 US$ = NRs.114) and is highly skewed towards a few wells off CFUGs; the high-and-low average annual income of one-third of CFUGs in the sample ranges from NRs. 33,116 to NRs 502,363. Timber income and user's contribution constitute the most important sources of income, comprising 40% and 25% respectively. The rural development investments of CFUG income are also highly variable and are shaped by income size of CF, and the other socio-political factors such as the number of households, distance to market, infrastructure status, and contextual factors. Overall, 44% of the CFUG income is invested in community development and 37% in forest conservation. Investment in community development increases with rising income. Accordingly, results presented here provide insights to promote community forests to generate more income which, indeed, could be a vehicle for community development as it appears in the mid-hills of Nepal.

**Keywords:** community forest; forest user group; annual income; investment; community development

---

## 1. Introduction

The share of forests in developing countries managed under community-based approaches is increasing and is currently estimated at around 732 million hectares, 28% of the world's forests representing 62 countries [1]. In Nepal, Community Forestry (CF) has about four decades of history which was initiated especially after the enactment of the National Forestry Plan in 1976. Initially, CF was started for reversing the deforestation and fulfilling the basic forest product needs of local people [2]. Now it becomes one of the dominant forest management regimes of Nepal that has been claimed as a vehicle of forest conservation as well as rural development. There are 22,266 CFs managing a total area of about 22.37 million ha involving almost 2.9 million households [3]. Conserving forest contributes to enhancing the welfare of the communities living close to forest [4] and contributes to poverty reduction in developing countries [5]. CF demonstrated success in its dual objectives of ecological restoration [2,6–8] and livelihood improvement through income generation and community development [1,9–17].

Forest act, 1993 defines CF as an autonomous body [18] that can develop, conserve, use and manage the forest and sell and distribute the forest products independently by fixing their prices

according to operational plan [19,20]. The community forest user groups (CFUGs) have legal right to collect, retain and redistribute forest revenue of products from community forests. Community Forest Development Guidelines, (2014) direct each CFUG to develop its own constitution and management plan [21]. In addition, this guideline also includes the mandatory provisions on investment of revenue generated, such as at least 35% of its income should be invested on pro-poor activities, and 40% in forest community development [21]. Based on legal and policy provisions, CFUGs are obtaining income from different sources including but not limited to the forest product sale, membership fee, penalty, awards and donation from governmental and non-governmental organizations. However, these policy provisions do not recognize the variability of CFs in Nepal. CF in Nepal not only varies between the different landscapes but also within the landscape. There are substantial differences between the CFUGs in terms of household size, forest area, species composition, accessibility to market, rural-urban context and dependency of community on forest resources. Similarly, CFUGs income varies by a socio-economic group [22,23] and is affected by several factors, such as location, species composition, and nature of the forest [24,25]. Despite regulating through blanket policy approach these differences have significant meaning and play a vital role in the income and investment pattern of the CFs. Thus, CFs are heterogeneous not only in terms of the amount of income they generate but also variation is seen on fund generation, mobilization and decision making regarding the fund mobilization.

CFUGs are being developed as the institution with the potentiality of public financing for infrastructure, forest management and other public services [11,26]. CFUGs fund has an important role in community development [16]. The income distribution and public service financing of CFs are highly skewed and largely depends on the presence of high-value timber species, the size of the CFUG membership, and the age of the CFUG [11]. Investment in public services is substantially determined by the income pattern of the CFUGs and is furthermore shaped by management costs and socio-political and contextual factors, such as the number of the user households in CFs, market distance and donor support in the CFUGs. Investment of CFUGs in private activities contributes more to household well-being than the investment in public goods [24]. Studies regarding the benefit of CF were more focused on the household level benefit analysis and have given little emphasis on the allocation of CFUGs fund to public goods.

With the initiation of active management of forest and commercialization of forest products, the income of the CFUGs is increasing in Nepal. However, it lacks systematic accounting at the national and even the district level. At the CFUGs level, poor record keeping, irregular audit and lack of user's knowledge on the financial management of CF fund have also posed difficulties in record keeping and data analysis. How the fund is utilized in different areas such as forest management, community development, poverty reduction is more important [24] than just how much CFUGs earns. Studies related to income and expenditure of CFUGs in Nepal are limited and confined only in a small number of CFUGs due to which there remain gaps on what is the status of CFUGs fund mobilizations, investment patterns and inherent problems therein. Over the past few decades, the concern on the contribution and distribution of revenue generated from the CFUGs has been increasing and there are few studies [11,24,26] that have contributed to quantify it. However, very few studies have explored the trend in CFUGs revenue and expenditure patterns over a time period, implying that our knowledge often rests on segmented data and snap-shots [19]. Investment in public goods by CFUGs is shaped by different factors associated with CF and its stakeholders but this aspect is little studied in the past. Most of the studies lack time series data or details beyond reporting income and expenditure. Finally, empirical studies on patterns of public income and expenditure are limited. Chhetri et al. [11] provide evidence on public finance potential of CF of mid-hills with details on income and expenditure pattern. However, it has not captured the time series data and lacks details on income and expenditure trends. Thus, this study aims to fulfill such gaps which will have implication in community forestry policies in the changing governance structure of the country.

This study contributes to our knowledge about income and expenditure size, its pattern and trend in community forestry of Nepal and the details on the investment pattern on rural development activities and the factors affecting the investment in rural community development. The analysis is based on a comprehensive data set on CFUG income and expenditures from a random sample of 43 mid-hill CFUGs in Kaski district, Nepal. The study provides new empirical insights into the community forestry financial potential. Furthermore, the study presents theory-led regression analyses of factors affecting CFUG expenditures that finance community development at the local level.

## 2. Materials and Methods

### 2.1. Study Area

The study was carried out in Kaski district that forms part of the mid-hills in the Gandaki province. Kaski district lies between 28°06′ and 28°36′ N latitude and 83°40′ to 84°12′ E longitude. Altitude varies across the district from 450 m in the south to 8091 m in the north. The area of the district is 2017 km². The lower elevated part of the district has a sub-tropical climate; climatic variations from sub-tropical through temperate, alpine and tundra are found across the district, south to north (1500 m to above 4500 m). Due to variations in the geographic and climatic zone in the district, there is variation in availability of vegetation too i.e., sub-tropical broad-leafed forest, temperate forest, sub-alpine forest, and alpine forest [27].

According to the population census 2011, the total population of the district is 492,098 out of which 236,385 are male and 255,713 are female whereas the sex ratio is 92.44 [28]. The population growth rate is 2.57, total household of the district is 125,673 [28]. Average household size is 3.92. Households headed by a male is 77,090 and that headed by females is 48,583 [28]. The district was chosen purposively to represent the socio-economic conditions in the region and representing mid-hill areas. The CFUGs with above five years of age and with updated audit report were chosen for the study. A total of 43 CFUGs were surveyed, randomly selected from CFUGs having an average annual income of more than NRs 10,000. The selected sites were verified and updated with the Division Forest Office (DFO) staff prior to the selection. The district has comparatively higher numbers of CFUGs handed over i.e., a total of 508 CFUGs and has variation in caste and ethnicity and income and expenditure ratio of the CFUGs.

Out of total forest area in district, 31% (28,575 ha) is under the jurisdiction of DFO and 69% (65,074 ha) is under Annapurna Conservation Area Project (ACAP). Overall, 68.22 % (19,495 ha) of the total potential community forestry area in Kaski has been handed over and benefited households are 46,692, implying ample of the forest is under community management [27]. Figure 1 shows the CFUGs selected for study in the Kaski district, which reveal that implementation, has focused on the lower elevation areas, a majority of the higher areas are managed under ACAP. The results in this paper are thus the only representative of the lower mid-hills in Nepal.

### 2.2. Data Collection and Analysis

The discussion was carried out with the officials of DFO and the district chapter of Federation of Community Forest User Groups, Nepal to obtain general information about fund generation and utilization pattern of CFUGs in this district. We reviewed the annual monitoring and evaluation reports of CFUGs published by DFO Kaski which gave the general overview of income and expenditure of CFUGs in the whole district. Preliminary information about the CFUGs and their fund were collected from the operational plan, constitution, and annual progress report and climate change adaptation plan of these CFUGs which have provisions about the fund generation and utilization of CFUGs. Account records of CFUGs of three fiscal years 2013/2014, 2014/2015 and 2015/2016 were reviewed and detailed information about their fund was obtained. CFUGs have to submit the audit report to Division Forest Office within 2 months of the end of the fiscal year. We also reviewed the audit report to validate the information collected from the CFUGs account record. Along with the account

information, attributes of the all studied CFUGs *viz.* market distance, house size, forest management status, forest type, and forest area were recorded. Information collected was verified by discussing with the executive committee members especially the president, secretary, and treasurer of CFUGs. Similarly, a details survey with all 43 CFUGs were carried out with pre-tested checklist/questioners attached to the financial recording format. The recorded information was validated during group discussions in a larger group of key informants—usually encompassing both present and past CFUG executive committee members. Data were collected from January to April 2016. For the analysis purpose, the income of CFUGs was categorized as wood income, non-wood income, DFO and donor support, user's contribution, Income Generating Activities (IGAs) activities, and last year balance. Likewise, expenditure was categorized as forest development, community development, training, administration, and interest/donation/prize. CFUGs were categorized into three income categories *viz.* low income (<NRs. 50,000), medium income (NRs. 50,000–NRs. 300,000) and high income (>NRs. 300,000). A linear regression was carried out to show the relationship between the CFUGs attributes and their investment in community development.

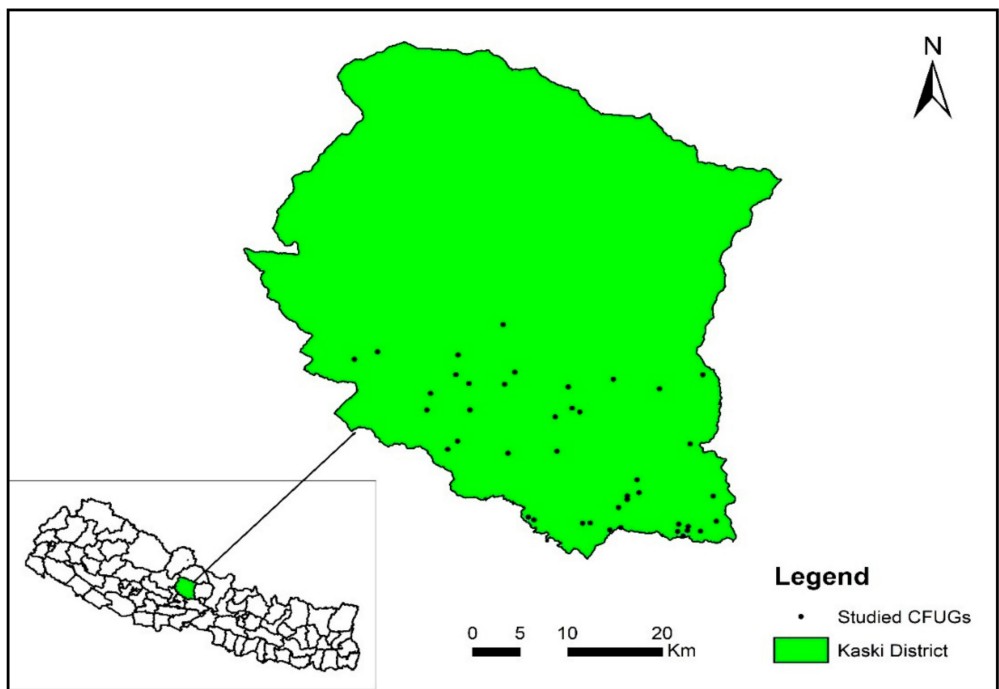

**Figure 1.** Map of Nepal with Kaski district and studied Community Forestry User Groups (CFUGs).

Investment in community development = f (income, number of households, market distance, donor support, infrastructure availability, women in the executive committee, Dalit (Dalit refers to a group of people who are religiously, socially, culturally, economically and historically excluded and treated as untouchables. They are natural resource dependent groups but comparatively have weak access in decision making.) in executive committee).

The mean and standard deviation and the expected signs of the independent variables and the explanatory variables is presented in Table 1.

**Table 1.** Descriptive statistics of the three-community development expenditure model variables (n = 43).

| Mode Variables | Mean | Standard Deviation | Expected Signs | Remarks |
|---|---|---|---|---|
| **Independent Variables** | | | | |
| Log of investment in community development in 2013/2014 | 4.13 | 5.25 | + | Model 1 |
| Log of investment in community development in 2014/2015 | 5.63 | 5.73 | + | Model 2 |
| Log of investment in community development in 2015/2016 | 6.40 | 5.44 | + | Model 3 |
| Log of average of the investment in community development of year 2013/2014; 2014/2015 and 2015/2016) | 8.30 * | 4.13 | | Model 4 |
| **Explanatory Variables** | | | | |
| Log of income in 2013/2014 | 9.49 | 4.71 | + | For Model 1 |
| Log of income in 2014/2015 | 11.39 | 2.24 | + | For Model 2 |
| Log of income in 2015/2016 | 10.68 | 3.28 | + | For Model 3 |
| Log of average income of year 2013/2014; 2014/2015 and 2015/2016) | 11.61 * | 1.23 | | For model 4 |
| Number of households in the selected CFUGs | 156.58 | 205.31 | + | All Models |
| Nearest market distance from the CFUG (Dummy, 1 = far) | 0.53 | 0.50 | +/− | All Models |
| Donor support to the CFUG in the last five year (dummy, 1 = yes) | 0.31 | 0.47 | − | All Models |
| Status of available infrastructure in the village (dummy, 1 = good) | 0.33 | 0.47 | − | All Models |
| Woman proportion in executive committee (EC) of the CFUG | 0.42 | 0.12 | +/− | All Models |
| Dalit Proportion in executive committee (EC) of the CFUG | 0.11 | 0.06 | +/− | All Models |

\* Some of the CFUGs have no investment in community development in some year and this has resulted in the higher average of three years investment in community development and similarly, some of the CFUGs have no income in some years which has resulted in the higher average income of three years.

## 3. Results

### 3.1. CFUG Income and Expenditure Pattern

The Average annual CFUG income in fiscal years year 2013/2014, 2014/2015 and 2015/2016 was found NRs. 169,050, NRs. 248663 and NRs. 230,961 respectively (Table 2) and overall average annual income were found NRs. 216,225. Highest mean income was obtained from wood in all three years and this covered about 40% of the total annual average income in all three years. User's contribution including membership fee, new member fee, penalty, user's support, interest application, and other contribution followed the wood income as CFUG income source (Figure 2). User's contribution was 20% to 34% of the total annual average income.

IGAs was also one of the important income sources that contributed 15% in 2013/2014, 24% in 2014/2015 and 16% in 2015/2016 of total annual average income. Income from non-wood income was found negligible. Similarly, we observed the last year balance in the CFUGs fund. The income from wood, user's contribution, and IGAs appear fluctuating.

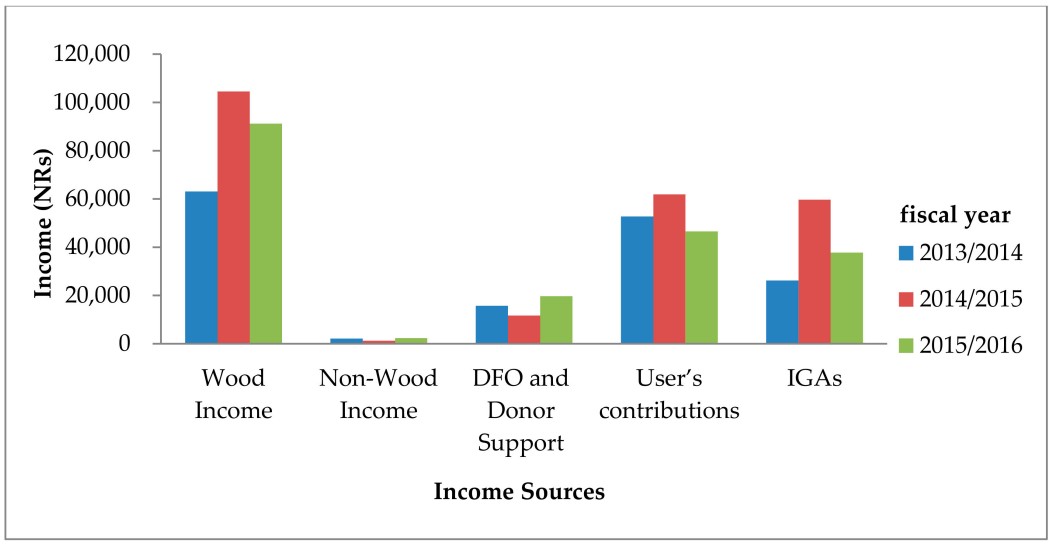

**Figure 2.** Key income sources of CFUGs in fiscal year 2013/2014, 2014/2015 and 2015/2016.

**Table 2.** Average annual CFUG income * (NRs.) divided by year and source (n = 43).

| CFUG Income | | 2013/2014 | | 2014/2015 | | 2015/2016 | |
|---|---|---|---|---|---|---|---|
| | | **Mean** | **SE** | **Mean** | **SE** | **Mean** | **SE** |
| Wood income | Timber | 47,595 | 26,880 | 78,120 | 34,371 | 72,777 | 29,215 |
| | Fuelwood | 15,353 | 8204 | 26,251 | 8842 | 18,226 | 9298 |
| | Pole | 118 | 97 | 170 | 162 | 189 | 163 |
| | Sub-total | 63,065 | 28,786 | 104,542 | 34,254 | 91,192 | 29,795 |
| Non-wood income | Fodder grass | 1534 | 897 | 740 | 382 | 1047 | 619 |
| | Thatch grass | 27 | 27 | 0 | 0 | 29 | 29 |
| | NTFPs | 312 | 231 | 109 | 77 | 1182 | 950 |
| | Others | 240 | 136 | 386 | 361 | 51 | 36 |
| | Sub-total | 2112 | 923 | 1235 | 537 | 2309 | 1178 |
| DFO and donor | Sub-total | 15,735 | 7735 | 11,705 | 5295 | 19,681 | 11,095 |
| User's contributions | Membership fee | 6489 | 1881 | 8922 | 2693 | 6891 | 2069 |
| | New member fee | 3810 | 2249 | 2131 | 1174 | 2089 | 1448 |
| | Penalty | 1990 | 1047 | 210 | 117 | 574 | 324 |
| | Users support | 7571 | 3170 | 9926 | 6135 | 9588 | 3960 |
| | Interest | 19,301 | 6288 | 15,748 | 3890 | 15,523 | 4101 |
| | Application | 1639 | 724 | 4747 | 3436 | 4677 | 3782 |
| | Others | 11,964 | 6412 | 20,230 | 8970 | 7250 | 2478 |
| | Sub-total | 52,764 | 10,530 | 61,914 | 12,579 | 46,592 | 8307 |
| IGA activities | Sub-total | 26,163 | 11,733 | 59,727 | 22,215 | 37,791 | 18,651 |
| Last year balance | Sub-total | 9211 | 6197 | 9539 | 7094 | 33,397 | 21,646 |
| Total | | 169,050 | 35,649 | 248,663 | 51,619 | 230,961 | 60,743 |

* Including last year's balance.

The result shows in average out of total income more than 75% has been invested by CFUGs for different activities each year. The three years average shows the share of expenditure is more in community development and forest development activities, which accounts on average more than 80% of total expenditure (Table 3; Figure 3). The key forest development activities include protection, NTFPs promotion, silvicultural operation, plantation, fire line construction, seedling production, and other various forest-related activities. Community development activities varied in different CFUGs according to their needs but the shared community development activities were electricity, community building, road/foot trail, education (Salary for school teachers), school building, scholarship, drinking water, health/sanitation, etc. Within the community development activities, the highest priority was given to electricity (40.31%) followed by community building (21.66%) and road/foot trail (10.77%).

Year wise analysis shows CFUGs invested the highest amount of fund in forest development (52%) followed by community development (26%) in the year 2013/2014 (Figure 3). However, the expenditure scenario changed in year 2014/2015 and 2015/2016 in which years priority was given to community development and the investment in community development works accounts for 52% and 49% of total expenditure respectively. Expenditure on training (1–2%), administration costs (14–15%) and interest/donation/prize (4–5%) remained consistent in these years. The average mean annual investment of CFUGs in community development works (44%) exceeds the forest development works (37%). The expenditure on administration, interest/donation/prize and training was found 14%, 4% and 1% of total expenditure respectively.

The average administration cost of CFUGs was found no more than 14%. There are a lot of activities requiring the budgetary expenses viz. staff, office rent, communication/phone, stationery, meeting, assembly, ranger service charge, Federation of Community Forest Users, Nepal (FECOFUN) charge, audit, and other miscellaneous expenses. Although in a small amount, CFUGs have also spent its income on interest, donation and prize distribution.

In absolute term, the result shows substantial variation in low, medium and high-income CFUGs and the amount of expenditure (Table 4). However, there is not much deviation in terms of the relative amount of expenditure of these groups of CFUGs. The average income of low-income CFUGs is six and half times lower than the total average income of CFUGs. Similarly, it is one and half times

lower in case of middle-income CFUGs, while it is twofold larger in case of high-income CFUGs. The average expenditure in all CFUGs category varies from 70 to 77% of total income of CFUGs (overall average is 76%) (Table 4), however, in absolute term, the higher income group have a higher amount of expenditure. The higher income CFUGs expenditure is double of the total average CFUGs expenditure while it is seven-fold lower in case of low-income CFUGs.

**Table 3.** Average annual CFUG expenditure (NRs.) by title and year (n = 43).

| CFUG Expenditure | 2013/2014 | | 2014/2015 | | 2015/2016 | |
|---|---|---|---|---|---|---|
| | Mean | SE | Mean | SE | Mean | SE |
| **Forest development** | | | | | | |
| Protection | 15,235 | 4704 | 22,504 | 8631 | 17,947 | 6010 |
| NTFPs promotion | 1821 | 1821 | 2809 | 1963 | 247 | 247 |
| Silvicultural operation | 33,041 | 12,145 | 23,327 | 8466 | 20,807 | 6287 |
| Plantation | 25,355 | 22,826 | 3264 | 1982 | 2721 | 1062 |
| Fire line construction | 0 | 0 | 6034 | 5687 | 1607 | 1199 |
| Seedling production | 382 | 350 | 1609 | 1609 | 2943 | 2943 |
| Others | 1068 | 638 | 931 | 524 | 70 | 70 |
| Sub-total | 76,902 | 25,983 | 60,479 | 15,641 | 46,342 | 10,746 |
| **Community development** | | | | | | |
| Education (School teacher) | 4113 | 4113 | 5649 | 5237 | 8137 | 5923 |
| School building | 2907 | 1832 | 3723 | 2652 | 4419 | 2710 |
| Scholarship | 84 | 59 | 0 | 0 | 31 | 31 |
| Drinking water | 70 | 70 | 9395 | 4825 | 9965 | 5549 |
| Road/ foot trail | 929 | 560 | 7699 | 4901 | 14,451 | 7935 |
| Electricity | 0 | 0 | 62,752 | 29,024 | 23,629 | 12,097 |
| Community building | 24,765 | 13,363 | 12,598 | 10,504 | 9046 | 3799 |
| Health/sanitation | 4107 | 2560 | 1690 | 1073 | 1718 | 814 |
| Others | 587 | 455 | 458 | 311 | 1379 | 1128 |
| Sub-total | 37,561 | 14,630 | 103,965 | 32,391 | 72,775 | 18,794 |
| **Training** | | | | | | |
| Training/workshop and tour | 0 | 0 | 850 | 850 | 0 | 0 |
| Forest management | 565 | 565 | 0 | 0 | 703 | 703 |
| skill development | 1746 | 1237 | 493 | 493 | 225 | 225 |
| Awareness | 557 | 400 | 214 | 173 | 0 | 0 |
| Others | 41 | 41 | 1358 | 1264 | 0 | 0 |
| Sub-total | 2910 | 1691 | 2915 | 1583 | 928 | 733 |
| **Administration** | | | | | | |
| Staff | 667 | 417 | 327 | 187 | 414 | 232 |
| Office rent | 1202 | 752 | 556 | 421 | 1251 | 637 |
| Communication/phone | 1996 | 785 | 3325 | 1685 | 2189 | 1099 |
| Stationery | 2892 | 1476 | 3000 | 795 | 1804 | 388 |
| Meeting | 1715 | 581 | 2754 | 981 | 5320 | 2331 |
| Assembly | 6606 | 4890 | 3319 | 1020 | 1455 | 644 |
| Ranger service charge | 1400 | 570 | 1897 | 868 | 891 | 544 |
| FECOFUN charge | 123 | 65 | 380 | 186 | 70 | 49 |
| Audit | 2125 | 383 | 2535 | 376 | 3263 | 743 |
| Miscellaneous | 3642 | 1613 | 6804 | 2860 | 4400 | 2786 |
| Sub-total | 22,370 | 6617 | 24,896 | 4532 | 21,057 | 5413 |
| **Interest/donation/prize** | | | | | | |
| Interest (bank, loan) | 4353 | 3960 | 6640 | 4524 | 3176 | 3056 |
| Donation | 1051 | 560 | 186 | 134 | 116 | 116 |
| Prize | 1498 | 1123 | 1030 | 494 | 2557 | 1998 |
| Sub-total | 6902 | 4431 | 7856 | 4509 | 5849 | 3602 |
| Grand Total | 146,644 | 37,436 | 200,111 | 46,396 | 146,951 | 30,629 |

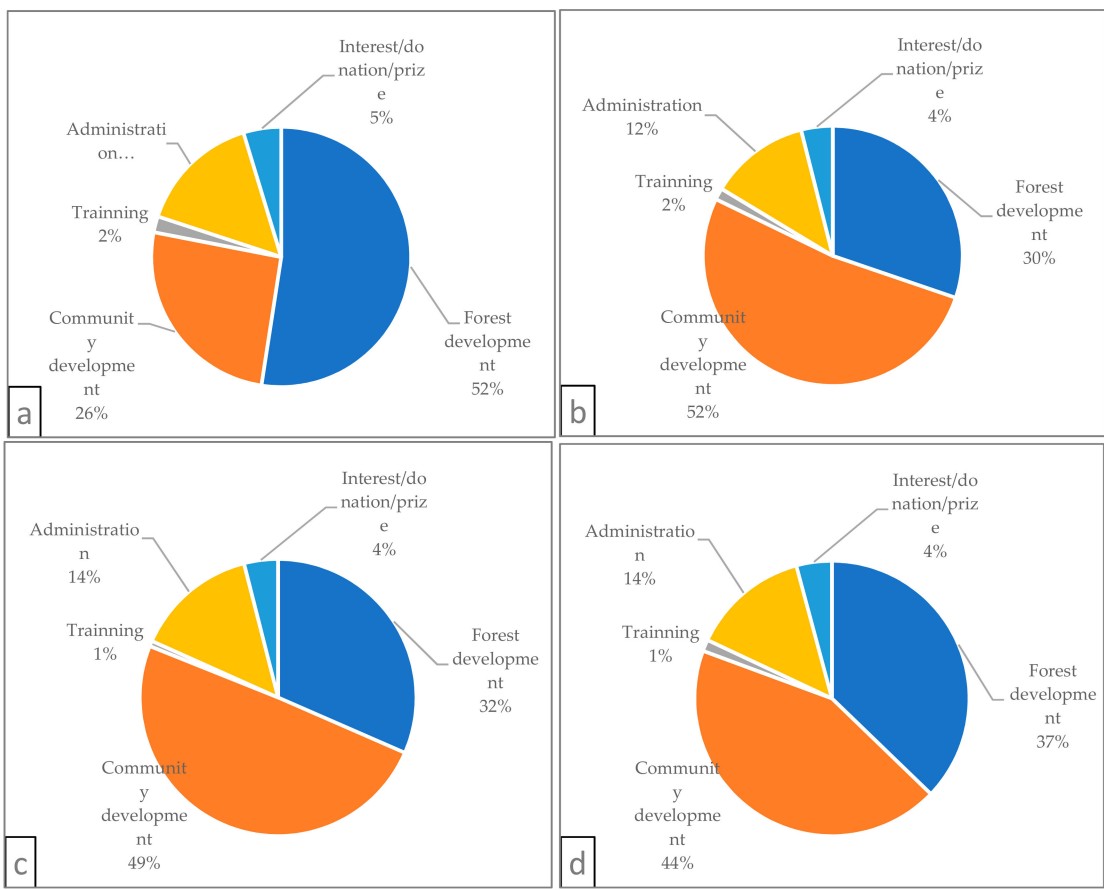

**Figure 3.** Expenditure pattern of CFUGs 2013/2014 (**a**), 2014/2015 (**b**), 2015/2016 (**c**), and average of all years (**d**).

**Table 4.** Average annual CFUGs income and expenditure by income group.

| Average Income | Income Group | | | All |
|---|---|---|---|---|
| | Low | Middle | High | |
| 2013/2014 | 30,307 | 129,765 | 356,989 | 169,050 |
| 2014/2015 | 39,428 | 136,271 | 585,235 | 248,663 |
| 2015/2016 | 29,613 | 112,787 | 564,865 | 230,961 |
| Overall average | 33,116 | 126,274 | 502,363 | 216,225 |
| Average expenditure | | | | |
| 2013/2014 | 18,204 | 57,285 | 373,617 | 146,644 |
| 2014/2015 | 25,593 | 120,021 | 467,184 | 200,111 |
| 2015/2016 | 25,379 | 98,876 | 325,281 | 146,951 |
| Overall average | 23,059 | 92,061 | 388,694 | 164,569 |
| Expenditure (% of total income) | 70% | 73% | 77% | 76% |

*3.2. Factors Affecting Investment in Community Development Activities*

The regression results showed that the effect of income on investment in community development activities is highly significant (*P*-value < 0.05): a 10% increase in income increases investment in community development activities by approximately 4.35%, 12.28% and 8.98% for the year 2013/2014, 2014/2015 and 2015/2016 respectively (Table 5). The number of households has also a positive impact on investment in community development which is significant for the year 2013/2014. Regarding market distance, different results were obtained for different years. In year 2013/2014 community development investment is found negatively correlated while for the year 2014/2015 and 2015/2016 it is found positively correlated. For year 2014/2015 relation of market distance and investment in

community development found significantly positively correlated. The relation between donor support and investment in community development activities was found negatively significant causing 43.15% decrease in investments by CFUGs in donor-supported CFUGs. Dalit representation in the executive committee was found significantly correlated with investment in community development activities for year 2014/2015 and for three years average (Table 5). The effect of infrastructure condition and women representation on EC is not significant in the case of individual years however it is significant for three years average. Similarly, the condition of infrastructure on villages has a negative impact on investment on community development activities as shown by a negative coefficient in all cases. The effect of women proportion in executive committee has found negative on community development activities investment for the year 2013/2014 and 2015/2016 while the effect is found positive for the year 2015/2016.

**Table 5.** Regression results of three different models for log of investments (in NRs.) in community development activities in three different years (n = 43).

| Independent Variables | 2013/2014 | | 2014/2015 | | 2015/2016 | | Three Years Average (2013/2014; 2014/2015; 2015/2016) | |
|---|---|---|---|---|---|---|---|---|
| | Co-eff. | *P*-Value | Co-eff. | *P*-Value | Co-eff. | *P*-Value | Co-eff. | *P*-Value |
| Constant | 1.171 | 0.780 | −18.159 | 0.007 | −3.535 | 0.483 | −18.639 | 0.002 |
| Log of income (Rs) | 0.435 * | 0.008 | 1.228 * | 0.002 | 0.898 * | 0.001 | 2.037 * | 0.003 |
| Number of households | 0.008 * | 0.050 | 0.002 | 0.658 | 0.002 | 0.603 | 0.003 | 0.214 |
| Market distance (1 = far) | −0.322 | 0.837 | 3.698 * | 0.039 | 0.517 | 0.761 | 0.005 | 0.506 |
| Donor support (1 = yes) | −4.315 * | 0.016 | 0.494 | 0.799 | 0.939 | 0.608 | −0.482 | 0.701 |
| Infrastructure (1 = good) | −0.703 | 0.673 | −1.442 | 0.446 | −0.816 | 0.651 | −2.212 * | 0.054 |
| Woman proportion in EC | −2.652 | 0.710 | 10.584 | 0.195 | −3.770 | 0.620 | 2.749 | 0.600 |
| Dalit Proportion in EC | 2.591 | 0.829 | 30.657 * | 0.027 | 13.078 | 0.307 | 19.765 * | 0.013 |
| Adjusted R-square | 0.2696 | | 0.2304 | | 0.2138 | | 0.4552 | |

\* Significant at 0.05 level of significance.

## 4. Discussion

### 4.1. CFUG Income and Expenditure

The average annual income of CFUGs was obtained NRs. 230,916 which is reasonably higher than the findings of other studies. Pokharel [23] studied 100 CFUGs of Kaski, Tanahu and Lamjung districts and reported the average annual income of CFUGs as NRs. 63,202. Equally, other studies from CFUGs in the mid-hills of Nepal also reported the lower average annual income. The average annual income of CFUGs was obtained as NRs. 22,000 [11] and 24,000 [26] which are far lower than the income of CFUGs found by this study. The biophysical setting of these districts is similar however there are several socio-economic changes and development happening in the last five years in the studied district. Likewise, evidence besides an increase in average income is that CFUGs are now gradually transforming from traditional protection-oriented forest management to active utilization of forest products. Income of CFUGs has been continuously increasing, for instance [29] reported around the seven-fold increase in average income of CFUGs in the six-year period. Likewise, a gradual increase in CFUGs income was also confirmed by other scholars from Nepal's study [13]. The high surge of income from wood in later fiscal years was observed showing the higher harvesting and commercialization of timber and other wood from the community forest area. After formulating the Scientific Forest Management Guideline 2014, the scientific forestry has taken momentum in the collaborative forest and in mid-hills community forest area including Kaski district [30]. This demands active forest management and is financially attractive [31]. However, the income from the wood is grounded on the annual allowable cut that is prescribed on forest operational plan of CFUGs. Equally, the income from the wood might increase in the future as recent policies and programs of government have emphasized scientific forest management that allows CFUGs to harvest all mature trees. The CFUGs

mean average expenditure of three years in forest development and community development was found about 37% and 44% respectively. A similar finding was observed in previous studies in mid-hills of Nepal [26]. The major investment in community development activities incudes the basic need of rural people/community namely electricity, community building, road, school support and drinking water system (Figure 4). The expenditure pattern and priority were found to be different in different years. However, it seems that there is a shift towards rural development activities in recent years. For example priority was given for forest development in year 2013/14 by allocating 52% of total expenditure on these activities, while the priority was found shifting to community development later in two fiscal years 2014/15 and 2015/16 by allocating 52% and 49% of total expenditure respectively in community development activities and the studies in this region by [11,16] also have similar findings.

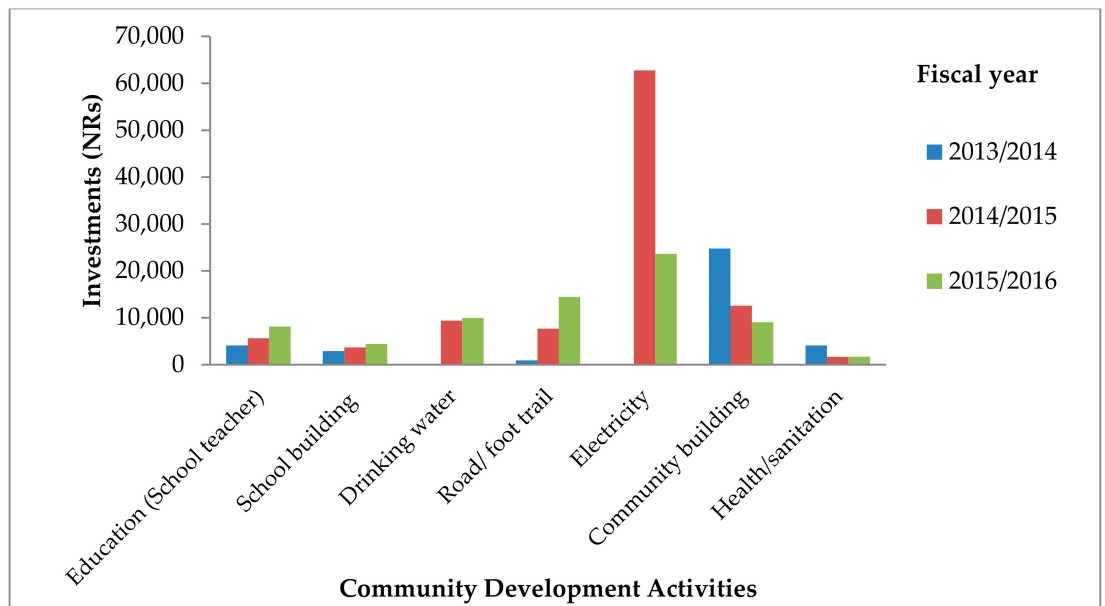

**Figure 4.** Investments pattern in different community development activities in fiscal year 2013/2014, 2014/2015 and 2015/2016.

In recent years the forest management has been shifting from protection-oriented management to production-oriented management system. Chhetri et al. [11] from their study on the community forests of Gorkha district reported that CFUGs spend 45.2% of their total expenditure on local public services and infrastructure and 46.6% on the forest management activities. Electricity was found as the focused community development work accounting 40% of total expenditure in our study. In the fiscal year 2013/2014, CFUGs did not allocate budget for electrification but in the next two fiscal years investment on electrification is high showing the CFUGs investment in basic rural need and emergency activities is high. Other community development activities with a significant amount are community building and road/foot trail construction.

The income category of CFUGs seems highly skewed towards a few wells off CFUGs. The high-and-low average annual income of one-third of CFUGs in the sample ranges from NRs. 33,116 to NRs 502,363. However, there is very minimum variation in terms of investment of income across the years. The substantial portion of the income has been invested and used for different forest and rural development activities. The investment pattern of CFUGs on community development and forest development activities have higher share and investment in administrative activities is minor. This shows the investment pattern of common funds of CFUGs seems highly deviated towards community and forest development. Though the relative investment ratio of different income group CFUGs is higher, in absolute term the higher income group CFUGs investment is significantly higher in community and forest development activities. The investment pattern of CFUGs looks

somehow sustainable as the investment pattern on administrative, forest development and community development seems quite usual, meaning that the investment in administrative activities is lower than other investments categories and all the categories is duly considered. However, it depends on several factors and drives through the income of the CFUGs and is highly skewed. This infers that additional support should be targeted to those low-income CFUGs so that they can enhance their income.

Nepal's forest policies in recent years are gradually incorporating the provisions to regulate the fund mobilization of CFUGs. Forest Act, 1993 and Forest Regulation, 1995 have limited provisions in their initial phase of implementation but in recent years these policies were amended to incorporate and regulate the fund mobilization in CFs. The Guidelines for the Community Forestry Development Program (Third Revision, 2014) states that the community forest user groups should spend at least 25% of their income in the forest conservation, management and utilization and 40% in community development activities. The study suggests that the CFUGs have fulfilled the legal obligation and are towards more rural development focused.

*4.2. Community Development Investment Model*

We analyzed the impact or various factors affecting CFUGs investment in community development activities. Income of CFUGs has a positive significant impact on investment in community development which was also similar to findings of the previous study [11]. When the income of CFUGs increased then they could save a higher amount of fund even after investing for forest management and basic administrative activities which provide CFUGs ample amount for investing in community development. Lund [32] reported that surplus income from the use of natural resources is used to finance public services at the community level. The positive impact of a number of households on investment in community development has been observed in this study. Although the investment in public infrastructure does not necessarily fairly benefit to all households [24] with an increase in household size, there is a high demand for public infrastructure and community development.

The presence of good infrastructures in the community shows a reduction in the CFUGs investment in community development activities. Donor support in the particular community forestry was found negatively correlated with significant regression coefficient. In the case of Kaski district, donors like hariyo ban program and multi-stakeholder forestry program were working to support CFUGs. Donor support can change even the domestic community-based forestry policy of the aid recipient countries [33] and in this case, the domestic policy of CFUGs regarding the investment in community development was significantly changed due to donor support. In this study, Dalit representation in executive committee has found positive and significant impact in investment in community development. The similar result was reported by Pokharel [23] reporting the positive correlation between the representation of Dalits and financing in pro-poor development activities. The economic status of Dalit is normally lower and usually lives in marginalized areas of villages lacking the basic services. Dalit representative raises voice to develop infrastructure and facilities in villages so that they could have easy access to the public infrastructure services.

## 5. Conclusions

We found new evidence on the financing potential of community forestry in the mid-hills of Nepal based on the detail report and analysis of income and expenditure pattern of 43 randomly sampled CFUGs having an average annual income of more than NRs 10,000. Our sampled community forests were found unequal in terms of their assets, opportunities and their ability to generate income and redistribute the benefits. We found that a substantial portion of CFUG income depends on the wood-based income followed by the contribution of users in the form of membership fees and interest and is found highly skewed towards a few wells off CFUGs. This implies CFUGs with largest forest area and valuable timber species could be the focus while handing over new CFUGs or developing new forest management policies and interventions.

The significant amount of income of the CFUGs has been invested in community development activities and this is increasing in recent years. The CFUGs financing community development is given priority with due consideration in the cost of forest development and administration. This indicates the high potential of CFUGs to support rural development that is guided by collective decision and efforts. The findings that higher income CFUGs has higher and significant potential to invest in rural development activities indicates that the investment of CFs revenue in collective work for collective benefits is increasing and this is a positive sign for rural development. Though the larger household coverage of CFUGs and inclusive executive committee have increased the CFUGs investment in rural development, the donor-supported CFUGs shows reluctance in public investment. It suggests that the number of households and inclusive executive committee formation should be considered while establishing new CFUGs. The finding that financing on community development is significantly reduced in CFUGs with low income suggests the need for due attention to improving income condition of CFUGs. The results presented in this paper, and the differences in findings between this and other studies on CFUG income, warrant research for more detailed studies based on larger samples of randomly chosen CFUGs that look beyond averages to gauge income-generating potential of CFUGs and redistribution of the income in community development in Nepal.

**Author Contributions:** Conceptualization, methodology, formal analysis, investigation, writing—original draft preparation, P.K.C.B., P.B., and B.B.K.C., Writing—review and editing, P.B., B.B.K.C., G.P., and P.K.C.B., Supervision, B.B.K.C., and C.P.U., Funding acquisition, P.K.C.B.

**Funding:** This research was partially funded by Science and Power in Participatory Forestry (SCIFOR) project funded by Danish Consultative Research Committee on Development (DANIDA) and Nityananda Service Foundation, Pokhara.

**Acknowledgments:** We would like to put sincere thanks to the Community Forest User Groups who helped us during the fieldwork and who always welcomed us with a smile. Thanks to Division Forest Office, Kaski, Dhurba Bahadur Malla, Thakur Giri, Sami Shrestha and Bishnu Paudel for helping us in the field work.

**Conflicts of Interest:** The authors declare no conflict of interest.

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
