# Peer review of "Importance of Community Forestry Funds for Rural Development in Nepal"

_resources, doi:10.3390/resources8020085_

Round 1
Reviewer 1 Report
Dear authors,
The following comments are the result of my review of your manuscript “Importance of Community Forestry Funds for Rural Development in Nepal”. I will first give you some general remarks, followed by specific remarks according to the line numbers in the manuscript.
General remarks:
A language check is required. The quality is poor. There are dozens of spelling mistakes, is/are used wrong, nouns instead of verbs, …
Inconsistent handling of “we “ and “our …” (Anglo-American style)
Number formatting: Sometimes 1,000 sometimes 1000; Inconsistent number decimal numbers; Especially in all tables!
Currency: Values are given in NRs; For an international audience it might be easier to understand in $; somewhere in the script you present the exchange rate. This should at least be presented in the beginning.
Monetary values: There is no clue if you made an inflation adjustment – as far as I found out, Nepal has an annual inflation rate of ~ 10% which makes a big difference, at least when comparing to study results from 10 years ago
Units of values are often missing
Inconsistent use of NRs or Nrs, mid-hills or mid-Hills or mid hills
Figure 4 is not mentioned in the text --> remove it if it is not required
There are some errors with the sources: Some seem to use different style and therefore are missing in the reference list
Is donor support the same as subsidy? You should explain who the donors are as it seems to be a significant factor in your regression
You show that the main source of income for the CF is the timber production and mention that they are transforming from conservation of forest land to wood production.
Is there any system to check for the sustainability of timber production (e.g. forest management plans, certificates like PEFC/FSC, …)? What is the potential for the future? This is in my opinion a crucial and missing point in the discussion.
Specific remarks:
15: currency NRs vs. $ and number formatting 216,225
19: which “the other socio-political” factors?
34: replace “having” by managing
35: number format: 22.37
40: “… autonomous body ..”: I would suggest a source (e.g. law) for this
44: Source?
72,73: What is the meaning of the sentence starting with “Poor ..”?
74: Sentence “How the … earns” needs a source
103: add the word latitude after the coordinates;
103: number formatting: 8,091
103: Kaski district is part of mid-Hills and has an altitude below and above the values presented for the mid-Hills
104: km² instead of Sq. Km
108: broad leafed forest
Figure 1: Add a coordinate grid (then you can remove the coordinates in line 102 and 103);
110: source for the population census 2011 is missing
110, 111: number format: 4,92,098 – is this a formatting error or are the numbers wrong? Half a million of five million people?
111: sex ratio 92.44?
113: source?
122, 123: round the numbers
149: IGA is not explained until line 168
154: you should mention who the Dalit are and why it is important if there are Dalit in the executive committee
162: Nrs NRs used randomly; Number formatting, …
168: IGA already used in line 149 without explanation
171: I don´t see and “increasing trend” in Figure 2; Nevertheless, the income from Wood strongly depends from the level of actual cut which is very likely do vary between the single fiscal years! Furthermore does the timber price vary as well (at least in most economies). Conclusions about the trend based on 3 fiscal years might be misleading.
198: FECOFUN: The abbreviation is missing
Figure 3: The formatting of the figure needs a lot of modification. E.g. Administrati on … (value missing), word wrap, …
Figure 4: The whole figure is not mentioned in the text
206: low, medium and high income: What is the definition for the groups? Explain
211: “… 70 to 77% …” are these figures shown? Where do I find them?
221 and 226: the P-value can be found in Table 5; no need to mention them twice
234: “… (in Rs.) ..” is Rs NRs?
Table 5: You present values for the 3 fiscal years. Most independent variables seem to be significant in only a single year and are far away from significant in other years. I am really not sure if your conclusions based on the results are correct. I would at least suggest to test also with mean values of all 3 fiscal years and check which variables have significance.
241: The value 63,202 is from a 10 year old study. With an annual inflation of 10% in Nepal the valorized value is 165,000 NRs which is not that far away from your study´s result
305: reduction instead of reduced (noun vs. verb)
Author Response
Response to Reviewer 1 Comments
General remarks:
A language check is required. The quality is poor. There are dozens of spelling mistakes, is/are used wrong, nouns instead of verbs,
We have carefully checked the language error and corrected accordingly. For instance: line 36. 49, 54, 56, 57, 58, 59 and up to the conclusion section
Inconsistent handling of “we “ and “our …” (Anglo-American style)
Checked and corrected
Number formatting: Sometimes 1,000 sometimes 1000; Inconsistent number decimal numbers; Especially in all tables!
Checked and corrected
Currency: Values are given in NRs; For an international audience it might be easier to understand in $; somewhere in the script you present the exchange rate. This should at least be presented in the beginning.
Checked and corrected
Monetary values: There is no clue if you made an inflation adjustment – as far as I found out, Nepal has an annual inflation rate of ~ 10% which makes a big difference, at least when comparing to study results from 10 years ago
Suggestion acknowledged.
However, we think this is not much applicable in our case. It is forest resource-based income and the price of timber that CFUGs allocate has not much changed for the last five to ten years.
Units of values are often missing
Checked and corrected (particularly in case of a hectare (ha) and NRs)
Inconsistent use of NRs or Nrs, mid-hills or mid-Hills or mid hills
Checked and corrected
Figure 4 is not mentioned in the text --> remove it if it is not required
Checked and mentioned as needed
There are some errors with the sources: Some seem to use different style and therefore are missing in the reference list
Checked and corrected
Is donor support the same as subsidy? You should explain who the donors are as it seems to be a significant factor in your regression
Checked and corrected (line 338-340)
You show that the main source of income for the CF is the timber production and mention that they are transforming from conservation of forest land to wood production.
Timber has been the major source of income for CFUGs in Nepal. Normally, CFUGs are allowed to harvest dead, decay, deform, and diseased trees first and then only other green trees if the demand of users exceeded and if it is within the limit of the Annual Allowable Cut prescribed on operational plan. However, we have now mature forest and new policies have been focusing on active forest management meaning that CFUGs can harvest all mature trees and open canopy for regeneration. Thus this will increase the wood-based income.
Is there any system to check for the sustainability of timber production (e.g. forest management plans, certificates like PEFC/FSC, )? What is the potential for the future? This is in my opinion a crucial and missing point in the discussion.
Added in the discussion (line 310-315)
Specific remarks:
Reviewer comments | Response |
15: currency NRs vs. $ and number formatting 216,225 | The exchange rate of NRs and $ is provided and number formatting is revised |
19: which “the other socio-political” factors? | Other socio-political factors such as the number of households, distance to market infrastructure status, etc added in the text |
34: replace “having” by managing | Replaced |
35: number format: 22.37 | Corrected |
40: “… autonomous body ..”: I would suggest a source (e.g. law) for this | Forest Act, 1993 is cited |
44: Source? | Community Forest Development Guideline 2014 is cited |
72,73: What is the meaning of the sentence starting with “Poor ..”? | It means that record keeping is not systematic. This sentence is paraphrased |
74: Sentence “How the … earns” needs a source | Source is cited |
103: add the word latitude after the coordinates; | Added |
103: number formatting: 8,091 | Corrected |
103: Kaski district is part of mid-Hills and has an altitude below and above the values presented for the mid-Hills | The confusing sentence is deleted |
104: km² instead of Sq. Km | Corrected |
108: broad leafed forest | Corrected |
Figure 1: Add a coordinate grid (then you can remove the coordinates in line 102 and 103); | Coordinates were stated in the text |
110: source for the population census 2011 is missing | Cited |
110, 111: number format: 4,92,098 – is this a formatting error or are the numbers wrong? Half a million of five million people? | Corrected |
111: sex ratio 92.44? | The ratio of male and female population |
113: source? | Cited |
122, 123: round the numbers | Corrected |
149: IGA is not explained until line 168 | Explained |
154: you should mention who the Dalit are and why it is important if there are Dalit in the executive committee | Introduction and their importance in the executive committee is in footnote |
162: Nrs NRs used randomly; Number formatting, … | Corrected and made consistent |
168: IGA already used in line 149 without explanation | Corrected |
171: I don´t see and “increasing trend” in Figure 2; Nevertheless, the income from Wood strongly depends from the level of actual cut which is very likely do vary between the single fiscal years! Furthermore does the timber price vary as well (at least in most economies). Conclusions about the trend based on 3 fiscal years might be misleading. | Though the trend if fluctuating with an increasing trend, the income from the timber is the major income source. Yes, the timber income depends on the amount of the allowable cut prescribed on the management plan and this will increase if the forest condition gets better. In case of price, it doesn't very much for CFUGs however as CFUGs revise their operational plan in every 5 to 10 years, they do some changes based on the forest productivity. However, if we look at the timer price there is no much variation in price within CFUGs. |
198: FECOFUN: The abbreviation is missing | The full form is written |
Figure 3: The formatting of the figure needs a lot of modification. E.g. Administrati on … (value missing), word wrap, … | Figure formatting is corrected |
Figure 4: The whole figure is not mentioned in the text | It is now mentioned in the text |
206: low, medium and high income: What is the definition for the groups? Explain | Low income <NRs. 50,000, medium income (NRs. 50,000 – NRs. 300,000) and high income >NRs. 300,000 is explained in methods and materials part |
211: “… 70 to 77% …” are these figures shown? Where do I find them? | Table 4 |
221 and 226: the P-value can be found in Table 5; no need to mention them twice | Removed |
234: “… (in Rs.) ..” is Rs NRs? | Corrected NRs. |
Table 5: You present values for the 3 fiscal years. Most independent variables seem to be significant in only a single year and are far away from significant in other years. I am really not sure if your conclusions based on the results are correct. I would at least suggest to test also with mean values of all 3 fiscal years and check which variables have significance. | We have run the regression for the mean value of all three years and tested it, the results support the conclusion we have drawn. |
241: The value 63,202 is from a 10 year old study. With an annual inflation of 10% in Nepal the valorized value is 165,000 NRs which is not that far away from your study´s result | We have not considered inflation in case of the timber price in CFUGs. There is not much variation in the price of the timber in CFUGs and the harvest is done based on the operational plan which is the same for the first 5 to 10 years. |
305: reduction instead of reduced (noun vs. verb) | Corrected |

Reviewer 2 Report
It is necessary to strength the discusion and conclusions refering other studies of communities around the world. It is necessary to make a contrast among the results of this study with other cases outside Nepal. In sum, it is necessary to show similarities and contrasts in relation with other real cases around the world experiences.
In discussion and conclusions: Is this community development investment model a succesfull strategy? support it. If it is necessary, to establish new components for improving the sustainable forestry development in Nepal.
Therefore, I think it is necessary to support the research with more discussion and evidences in the international level supporting the conclusions. It is necessary to explain in more detail the connections and results of national and international cases related with the study objective. Also, they need to discuss the real impacts of this kind of actions with criteria of sustainable development.
Author Response
Response to Reviewer 2 Comments
Point 1: It is necessary to strength the discussion and conclusions referring other studies of communities around the world. It is necessary to make a contrast among the results of this study with other cases outside Nepal. In sum, it is necessary to show similarities and contrasts in relation with other real cases around the world experiences.
Response 1: We found very few studies outside Nepal on the investment pattern of commons fund however we have tried to add some study outside Nepal and made contrast and compare to strengthen the discussion and conclusion.
For instance:
Lund, J.F. Is small beautiful? Village level taxation of natural resources in Tanzania. Public Administration and Development 2007, 27, 307–318.
Rahman, M.S.; Sadath, M.N.; Giessen, L. Foreign donors driving policy change in recipient countries: Three decades of development aid towards community-based forest policy in Bangladesh. Forest Policy and Economics 2016, 68, 39–53.
Point 2: In discussion and conclusions: Is this community development investment model a successful strategy? Support it, If it is necessary, to establish new components for improving the sustainable forestry development in Nepal.
Response 2: We have added discussion accordingly and conclusion has been derived based on it (see line 311 to 317 and 361 to 365):
It looks somehow sustainable as the investment pattern on administrative, forest development and community development seems quite usual, meaning that the investment in administrative is lower than other investments categories. However it depends on several factors and drives through the income of the CFUGs and is highly skewed. Our result also supports this, the size of the CFUG and the forest type has strong connection with the income. Thus we have suggested considering the CFUGs with largest forest area and valuable timber species while handing over new CFUGs or developing new forest management policies and interventions.
Similarly, the additional support should be targeted to those low income CFUGs from other means so that they can enhance their income. As our results shows the high income CFUGs can sustain and low income CFUGs need additional support. Thus we suggest amending the current policy.
Point 3: Therefore, I think it is necessary to support the research with more discussion and evidences in the international level supporting the conclusions. It is necessary to explain in more detail the connections and results of national and international cases related with the study objective. Also, they need to discuss the real impacts of this kind of actions with criteria of sustainable development.
Response 3: - We have added some references and evidences on result and discussion portion to support the inference drawn in the conclusion.
We think the point raised concerning the discussion on sustainable development criteria is important however it is not quite directly associated with the scope of this study. Yes there are some avenues that income generated from CFUGs will continue in development of the country and it might trickle down and benefit others but we need further study for this.

Round 2
Reviewer 1 Report
Dear authors,
As far as I can see by your report, you used most of my comments/suggestions in the revised version of the manuscript.
I did not check the language and style of the revised manuscript, but it seems to me that you made some major corrections.
Referring to the methodical issues, I think that most important critical points are claryfied in the current version of the manuscript.